# Biochar-Derived Persistent Free Radicals: A Plethora of Environmental Applications in a Light and Shadows Scenario

**DOI:** 10.3390/toxics12040245

**Published:** 2024-03-27

**Authors:** Silvana Alfei, Omar Ginoble Pandoli

**Affiliations:** 1Department of Pharmacy (DIFAR), University of Genoa, Viale Cembrano 4, 16148 Genoa, Italy; omar.ginoblepandoli@unige.it; 2Department of Chemistry, Pontifical Catholic University, Rua Marquês de São Vincente 225, Rio de Janeiro 22451-900, Brazil

**Keywords:** biochar (BC), pyrolysis, biochar-derived permanent free radicals (PFRs), reactive oxygen species (ROS), PFR-mediated BC applications, environmental risk

## Abstract

Biochar (BC) is a carbonaceous material obtained by pyrolysis at 200–1000 °C in the limited presence of O_2_ from different vegetable and animal biomass feedstocks. BC has demonstrated great potential, mainly in environmental applications, due to its high sorption ability and persistent free radicals (PFRs) content. These characteristics enable BC to carry out the direct and PFRs-mediated removal/degradation of environmental organic and inorganic contaminants. The types of PFRs that are possibly present in BC depend mainly on the pyrolysis temperature and the kind of pristine biomass. Since they can also cause ecological and human damage, a systematic evaluation of the environmental behavior, risks, or management techniques of BC-derived PFRs is urgent. PFRs generally consist of a mixture of carbon- and oxygen-centered radicals and of oxygenated carbon-centered radicals, depending on the pyrolytic conditions. Here, to promote the more productive and beneficial use of BC and the related PFRs and to stimulate further studies to make them environmentally safer and less hazardous to humans, we have first reviewed the most common methods used to produce BC, its main environmental applications, and the primary mechanisms by which BC remove xenobiotics, as well as the reported mechanisms for PFR formation in BC. Secondly, we have discussed the environmental migration and transformation of PFRs; we have reported the main PFR-mediated application of BC to degrade inorganic and organic pollutants, the potential correlated environmental risks, and the possible strategies to limit them.

## 1. Introduction

Biochar (BC) is a stable carbon-rich black solid substance produced from vegetable or animal biomass feedstocks when pyrolyzed. Pyrolysis is a procedure that involves the heating of substrates at 200–1000 °C under oxygen-limited conditions [1]. The term “biochar” derives from the combination of “bio-,” which stands for “biomass,” and “char,” meaning “charcoal.” In recent years, BC has received widespread attention due to its potential application in carbon sequestration, soil amendment/remediation, wastewater treatment, and catalysis [2,3,4,5,6,7,8,9,10,11]. In particular, several ground-breaking studies have been carried out to investigate the potential of BC in alternative energy production and in the recovery of value-added chemicals/by-products [3]. In this regard, Lee et al. have used BC as briquettes and electrodes for microbial fuel cells (MFCs) finalized for alternative energy production [5]. Zhang et al. have employed BC as an additive/catalyst in anaerobic digestion and transesterification reactions for biogas [6], while Behera et al. employed BC to produce biodiesel [4]. Environmental applications of BC for reducing gaseous emissions, mainly by carbon and nitrogen sequestration, are also gaining attention [8].

BC has been applied to prevent eutrophication by recovering excessive nutrients, including nitrogen and phosphorus, from wastewater [11], while the agronomic application of BC, due to its characteristics of high cation exchange capacity (CEC) and specific surface area (SSA), have recently been reported by Zhao et al. [10]. Batch and column sorption experiments have shown that certain types of BC have good adsorption performance for heavy metals, dyes, or phosphate from aqueous solutions and are being investigated as cost-effective, promising, and eco-friendly alternative adsorbent materials [12]. Additionally, BC has demonstrated high efficiency in removing pharmaceuticals [13], pesticides [14], polycyclic aromatic hydrocarbons (PAHs) [15], and petroleum derivatives [16]. Also, the metal ion absorbent capacity of BC has been extensively reported in both the absence and presence of fulvic acid and humic acid [17]. As a soil improver, BC can reduce soil acidity and help maintain soil moisture and nutrient levels. Through its carbon sequestration action, BC performs climate restoration. Moreover, due to its strong adsorption capacity, BC can remove environmental xenobiotics, thus preventing their uptake in plants, animals, and humans [18,19,20,21]. Additionally, BC derived from the thermal treatment of organic material generally contains persistent free radicals (PFRs) bound to the external or internal surfaces of its solid particles [22,23]. Such BC-bound PFRs, which are reactive due to unpaired electrons, can persist for minutes and up to several months, in contrast to traditional transient radicals [24], thus conferring on BC the capacity to degrade organic pollutants through the generation of other reactive oxygen species (ROS) and sulfate radicals [25,26,27]. In this context, BC-bound PFRs have been investigated to activate persulfate (S_2_O_8_^2−^) to obtain sulfate radicals, which have efficiently degraded phenolic compounds and polychlorinated biphenyls [28], acid orange 7 [29], and sulfamethoxazole [30,31]. In the presence of PFRs, hydrogen peroxide (H_2_O_2_) or oxygen (O_2_) have been activated to produce hydroxyl radicals (OH•) and superoxide radicals (O_2_•^−^), which succeeded in efficiently degrading chloro-biphenyl [32], diethyl phthalate [33,34], and ciprofloxacin [35]. By the PFR-mediated activation of peroxyl mono sulfate (PMS), radical species such as SO_4_•^−^, •OH, and O_2_•^−^, as well as non-radical species such as ^1^O_2_ formed, which were the main contributors in the antibiotics’ degradation [36]. On the other hand, by stimulating the production of ROS, PFRs can inhibit seed germination and retard the growth of roots and shoots [37]. Additionally, BC production itself may cause the release of xenobiotics such as polycyclic aromatic hydrocarbons (PAHs), toxic inorganic elements, and dioxins, thus posing potential risks to human health and the environment [1]. The scientific community should evaluate BC and BC-bound PFRs’ positive and negative impacts before their extensive ecological application. Although the environmental behavior and risks of BC and BC-associated PRFs are increasingly attracting research attention, in the last ten years (the year 2024 excluded), studies into their toxicity remain very limited (96 publications), compared to those concerning PFRs in general (542 publications) and their degradation capability (402 publications) (Figure 1).

However, to safeguard the environment from BC-related PFRs’ potential adverse effects, it is necessary to comprehensively and systematically consider their environmental risks, formation mechanisms, and controlling factors [27], as well as the corresponding possible mitigation actions. Studies have shown that the type of biomass feedstock used to produce BC is pivotal in determining the physicochemical properties of the resulting BC, which also strongly affect the formation and characteristics of PFRs. To date, the types of biomasses used to prepare BC and involved in the investigation of BC-bound PFRs mainly include lignocellulosic biomasses (hemicellulose, cellulose, and lignin) from sources such as pine needles, wheat straw, lignin, cow manure, rice husk, and maize straw [38,39,40]. Additionally, cow dung (CD), sheep manure (SM), lotus stem (LS), and eggshell (ES) biomasses, representative of farm wastes, have been reported to provide BC containing PFRs [41]. Bamboo is an emerging starting material that is perfect for synthesizing BC and activated carbon (AC) due to its inexpensive cost, high biomass yield, and accelerated growth rate [42,43]. However, only a few researchers and scientists have used bamboo as a unique source for developing BC so far, as established by the number of publications on bamboo-derived BC developed in the last ten years (current year excluded) (331) vs. those on BC derived from different sources (13,630) (Figure 2).

Among the 331 publications on bamboo-derived BC, most were about their adsorption activity (119), followed by those on their degradation capacity (87). At the same time, few studies were conducted on their possible toxic action (Figure 3).

In this scenario, to promote the more productive and beneficial use of BC and the related PFRs and stimulate further studies to make them environmentally safer and less hazardous to humans, we have first reviewed the most common methods used to produce BC and its main environmental applications, as well as the reported mechanisms for PFR formation in BC. The main factors influencing the physicochemical properties of BC have also been reported. Secondly, we have discussed the environmental migration and transformation of PFRs, the main PFR-mediated applications of BC to remediate inorganic and organic pollutants described in the last five years, the correlated potential risks to the environment and humans, and the possible strategies to limit them. To confirm the relevance and essentiality of the present paper, a recent survey of the PubMed dataset has evidenced that, although the number of studies on BC-related PFRs has increased in the last few years, they are still very limited compared to those on BC in general (134 vs. 11646 from 2014 to the present). Additionally, the review articles on BC-associated PFRs that, by gathering information on the topic, could stimulate more research on it are indeed limited (16). Some recent examples could be considered works by Zhang et al., Liu et al., Luo et al., and Yi et al. [1,44,45,46]. In this scenario, this review can be considered essential because it offers an all-round and complete overview of both BC and BC-related PFRs, via an extensive discussion on both their beneficial impact and the possible risks to humans and the environment that could derive from their widespread and indiscriminate application. In the case of the present paper, we have provided a reader-friendly work where the information has mostly been organized into easy-to-read tables, schemes, and statistical graphs that could have a profound impact on its readers’ understanding.

## 2. Biochar (BC)

The constant growth in world population translates into a continued increase in the global energy requirement by all sectors and a dramatic decrease in fossil fuels, the primary energy source [47,48,49]. Furthermore, the effect of the resulting CO_2_ emissions on the environment determines additional global energy issues, which make the replacement of fossil fuels necessary and urgent [50]. In this regard, biochar (BC), mainly obtained from organic waste and possessing the capability of sequestering carbon, represents a rich carbon source and an alternative to fossil fuels [51,52]. Appendix A reports the biomasses commonly used for BC production [53,54,55,56,57,58,59,60,61,62,63,64,65,66].

BC obtained by the combustion of various biomasses, as reported in Appendix A, has been demonstrated to possess unparalleled physicochemical properties such as a large surface area, high porosity, the presence of several functional groups, high cation exchange capacity (CEC), long-term stability, etc. (Figure 4). Such properties make BC suitable for various applications, including, but not limited to, carbon sequestration, soil amendment, energy storage, and catalysis [67,68,69,70,71,72,73,74,75] (Appendix A). Additionally, BC is cost-effective, has an eco-friendly nature, and is endowed with reusability (Figure 4) [76,77]. Mainly, BC is increasingly gaining attention from many researchers as a material to efficiently remove various environmental contaminants, including antibiotics, thus reducing the emergence of microbial resistance [72,74].

Among the biomass waste materials appropriate for BC production, crop residues from agriculture, forestry, municipal solid waste, food, and animal manure have high potential [78,79,80,81,82,83].

### 2.1. Main Methods of Producing Biochar

As reported in the following Table 1, BC can be prepared rapidly using thermochemical conversion techniques such as pyrolysis, hydrothermal carbonization, gasification, flash carbonization, and torrefaction [84,85], with pyrolysis the most widely adopted (Section 2.1.1).

#### 2.1.1. Pyrolysis

Pyrolysis is a thermochemical process wherein the organic compounds present in the biomass are decomposed at a specific temperature [91]. Mainly, during pyrolysis, the thermal decomposition of organic materials occurs in an oxygen-free or oxygen-limited environment within a temperature range of 250–1000 °C [92]. In these conditions, the lignocellulosic components of biomass, such as cellulose, hemicellulose, and lignin, go through chemical reactions like depolymerization, fragmentation, and cross-linking, depending on the adopted temperatures. There are three principal possible products, including solid, liquid, and gas physical state materials. The solid products comprise BC and ash, while the liquid ones encompass bio-oils and tar, and the gaseous products (syngas) comprise carbon dioxide, carbon monoxide, hydrogen, and C1-C2 hydrocarbons [86]. As shown in Figure 5, during pyrolysis, the process parameters, including temperature, the type and nature of the biomass, residence time, heating rate, pressure, etc., could strongly affect BC yield and its physicochemical characteristics [93,94]. Moreover, although BC samples derived from different biomasses are all entirely made of carbon content and ash, their elemental composition, as well as their physical characteristics and properties, could differ enormously based on the type of biomass, reaction conditions, and type of reactor used during the carbonization process [95] (Figure 5). Consequently, every experimental condition and the starting raw material should be considered as a proof-of-concept of the future industrial application of BC.

The most widely used reactors for the chemical transformation of different biomasses include paddle kilns, bubbling fluidized beds, wagon reactors, tubular ovens, and agitated sand rotary kilns. However, temperature remains the primary operating process condition that governs the yield in BC, vs. those of the oily and gaseous products. Usually, BC yield decreases and syngas production increases when the pyrolysis temperature is improved [96]. Based on the heating rate, temperature, residence time, and pressure, pyrolysis can be categorized as fast or slow, as summarized in Appendix A [97]. Generally, fast pyrolysis is employed to maximize the liquid product yield, while slow pyrolysis is employed to maximize the solid product yield [98].

### 2.2. Biochar Characterization and Main Properties

The characterization of BC to determine its elemental composition is carried out by performing elemental analyses. Otherwise, its physicochemical, surface, and structural characterization is carried out by determining its surface functional groups, stability, and structure by employing various modern techniques reported in Appendix A [57].

As mentioned, the source of feedstock and the heat treatment temperatures during preparation are two significant factors that determine the physiochemical properties of BC.

The properties of pristine biomass that mainly influence the related BC include moisture content, ash content, calorific value, the percentage of lignin, cellulose, hemicellulose, fractions of fixed carbon, and volatile components [98]. High-yield BC with high porosity is achievable using biomasses possessing more lignin and less cellulose. Additionally, the volatile component, water content, particle size, and shape of the original biomass can also affect the properties of BC [98]. Table 2 reports the general chemical and physical features of BC, while Appendix A reports some characteristics of BCs produced from specific feedstocks at various production temperatures [99].

#### The Question of Temperature

As already reported, the pyrolysis temperature and feedstock greatly influence the physicochemical properties of BC, including pH, specific surface area, pore size, CEC, volatile matter, ash, and carbon content. CEC and volatile matter decrease with increasing pyrolysis temperature, whereas pH, specific surface area, ash, carbon content, and pore volume increase with an increase in pyrolysis temperature [100]. Increasing temperature also causes a decrease in the number of acidic functional groups, especially carboxylic functional groups, and causes the appearance of carbonylic functional groups and alkalinity [101]. In particular, unpaired negative charges forming during pyrolysis at higher temperatures enable BC to accept protons [101]. Although BC’s alkalinity increases with higher pyrolysis temperatures, thus improving its capacity to neutralize acids in soils, lower temperatures are necessary to preserve functional groups and obtain BC with higher CEC [102]. Low water content in BC, which reduces the possible microbial activity, promoting self-healing and degradation, is achievable at a higher temperature. However, the highly porous structure of BC obtained in such conditions causes the ready adsorption of moisture from the surroundings, thus increasing water content, re-enabling microbial activity, and contributing to self-heating and degradation [100].

During biomass decomposition to BC, the total surface area changes like the porosity due to the escaping of volatile gases and increases with increasing temperature [103]. In this regard, a large surface area affects CEC and water-holding capacity (WHC). Curiously, during pyrolysis, the hydro-properties of the initial biomass undergo several modifications depending on the pyrolysis temperature, which can translate into contradictory findings. Notably, with increasing temperature, due to a decrease in functional oxygenated groups and an increase in aromatic structure, the material’s affinity to water is altered, the hydrophobicity of BC becomes higher than that of pristine biomass, and its capacity to retain water will be lower. Conversely, thanks to increased porosity, which changes the amount of water that can be adsorbed, BC produced at high temperatures can hold more water in its porous structure than BC prepared at lower ones [104].

The mechanical stability of biomass usually decreases during pyrolysis and correlates inversely with the porosity and directly to the density of the BC and temperature. The electric conductivity increases with higher thermal treatment, improving the graphitic carbons’ crystallinity and the carbon-packed domains’ density [105]. BC with high mechanical stability can be produced from feedstocks with high density and lignin content, making lignin, the constituent, more resilient to decomposition and the loss of structural complexity. Conversely, BC with higher grindability can be obtained by the torrefaction of biomass with a larger amount of hemicellulose (e.g., agricultural residues) compared to woody biomass. The decomposition of biomass to BC causes a reduction in its bulk density and an increase in its porosity and, therefore, a decrease in its thermal conductivity, depending on the pyrolysis temperature. Concerning the electric properties of BC, the reduction in oxygenated functional groups and the appearance of conjugated double bonds cause an increase in conductivity and electromagnetic shielding efficiency, which make BC suitable as an additive in various composite materials (e.g., building materials such as cement). Furthermore, the effectiveness of shielding against electromagnetic interference is enhanced concerning the pristine biomass.

### 2.3. Possible Biochar Applications

The various properties of BC reported above, including its high carbon content, larger surface area, well-developed porous structure, and a surface sufficiently enriched with functional groups, render it potentially pertinent for various applications. In Table 3, we have reported the current possible environmental BC applications.

BC production could be an alternative to mitigate climate change by carbon sequestration in soil, thus retaining half of the carbon fixed in biomass during photosynthesis and reducing CO_2_, NO_2,_ and CH_4_ emissions [73]. Mainly, BC shows long-term stability in soil. The mean carbon residence time in BC has been estimated to be around 90–1600 years, depending upon the labile and intermediate stable carbon components [73]. Due to these characteristics, BC can sequester carbon in soil, thus decreasing carbon dioxide emissions into the atmosphere and those of nitrous oxide and methane by biotic and abiotic mechanisms [73]. Experiments have demonstrated that the emission of greenhouse gases (including CH_4_ and N_2_O) can be avoided by pyrolyzing waste biomasses [107]. Concurrently, the pyrolysis process balances fossil fuel consumption by producing bioenergy.

Interestingly, BC has been estimated to be capable of tackling 12% of the current anthropogenic carbon emissions. Furthermore, thanks to its high carbon content, BC can work as a soil conditioner, mainly by improving the soil’s physicochemical and biological properties. BC increases soil water retention capacity by ~18%, reduces nutrient leaching [68], and neutralizes acidic soils, thereby enhancing plant productivity, seed germination, plant growth, and crop yields. Additionally, wet BC prevents soil desiccation [68]. While it has been reported that soils treated with BC demonstrated improved microbial population and activity [108], null or positive effects were observed in the earthworm population in soils amended with wood-based BC [109].

The production of BC itself is an economical and mutually beneficial strategy to manage and eliminate waste from animals and plants and reduce the pollution associated with it [108]. Furthermore, when waste biomass that is derived mainly from animal manure and sewage sludge is pyrolyzed, the hazardous microbial population that is possibly present is killed, thus reducing its possible negative impact on the environment and humans. Unfortunately, toxic heavy metals from sewage sludge and municipal solid waste could persist in BC, which must be carefully checked and handled correctly before long-term soil application [110].

A remarkable potential use of BC, one that is still too little investigated and controversial, is the production of bioenergy as an alternative to fossil fuel that could lower carbon emissions. In this regard, while slow pyrolysis allows a lower yield of liquid fuel and more BC, fast pyrolysis provides more liquid fuel (bio-oil) and less BC [111]. Evidence has demonstrated that BC can be successfully applied in environmental remediation because it is capable of adsorbing both organic and inorganic contaminants, such as pesticides, herbicides, PAH, dyes, and antibiotics, as well as non-biodegradable metal ions, which are highly toxic to all living organisms [72,74]. BC can enhance the composting process by improving its physicochemical properties and microbial activities and promoting the decomposition of organic matter. Also, more investigations are needed to evaluate BC compost’s agricultural/environmental performance [69]. Table 4 summarizes some of the advantages and disadvantages associated with the production and use of BC.

As evidenced in Table 4, we benefit from additional advantages by producing BC from biomass, including waste biomass. The cost necessary to produce BC is six-fold lower than that of commercially available activated carbon (AC), which, unlike BC, is deprived of some properties of BC, such as its ion exchange capacity [112]. Generally, BC does not require further processing to be activated, and, thanks to its non-carbonized fraction and maintained oxygen-containing groups such as carboxyl, hydroxyl, and phenolic surface functional groups, BC is capable of adsorbing both organic as well as inorganic contaminants and of interacting with soil contaminants [72,74]. BCs produced from sewage sludge and manure have a high nutrient content for soils, thus enriching their quality [68]. However, apart from the advantages of using BC, there is a series of possible fallouts, as reported, that need consideration. Among these, the long-term removal of crop residues, like stems, leaves, and seed pods, for producing BC could reduce the overall soil health by diminishing the number of soil microorganisms and disrupting internal nutrient cycling, with a possible negative impact on soil biota, including short-term adverse effects on earthworm population density. In this scenario, there is a dire need for further extensive research so that any possible issues associated with its usage can be effectively resolved.

#### 2.3.1. Xenobiotics Removal by Biochar (BC)

As reported in the previous section, BC is a porous material, and its porosity, depending on the production temperature, allows it to interact with water nutrients and other materials, including inorganic metal cations and organic pollutants. Due to its enhanced porous structure, surface area, functional groups, and mineral components, BC is an optimal absorbent material for solutions. Although BC that is produced through pyrolysis has a relatively moderate adsorption capacity (3.6–6.3 g/g for BC prepared at a temperature range of 300–700 °C) [113], this can be enhanced by modifying its physicochemical properties through acid, alkali, or oxidizing treatments, while the surface area can be altered mainly using acid treatments [114,115,116]. As an adsorbent, BC can absorb organic and inorganic contaminants, such as PAH and phthalate acid esters, and its help in improving the treatment of sewage wastewater containing organic xenobiotics has been widely reported [117]. In this context, there are several main mechanisms used by BC for capturing inorganic or organic pollutants, which have been included and are discussed in Table 5.

Interestingly, BCs produced at higher temperatures exhibited higher sorption efficiency for the remediation of organic and metallic contaminants in soil and water. Additionally, it is worth mentioning that the sorption of organic xenobiotics by BC is more favorable than that of inorganic ones. Concerning complexation with metal cations, the smaller the ionic radius of metals, the greater the adsorption capacity by BC.

#### 2.3.2. Not Only Adsorption

It is commonly reported that the principal mechanism by which BC removes toxic heavy metals and other contaminants, including organic pollutants, is adsorption. Its adsorptive efficiency mainly depends on the type and number of functional groups, surface area, CEC, etc. However, previous research studies and reviews on BC have evidenced the presence, either on the surface or inside its particles, of free radicals known as persistent free radicals (PFRs), the nature of which depends strongly on the pyrolysis conditions and the formation and characteristics of which mainly differ based on the feedstock types. In this regard, several recent studies have mainly focused on the role of BC-related PFRs in the degradation of organic xenobiotics, in addition to their adsorptive capacity. Odinga et al., in their recent work, reviewed the application of BC-derived PFRs in environmental pollution remediation [27], while Fang et al. investigated the reactivity of PFRs in BC and their catalytic ability to activate persulfate to degrade pollutants [28]. However, Odinga et al. also considered and commented on the possible environmental risks of PFRs from BCs, which represent the shadows associated with these chemicals and need further study, knowledge, and regulation before their extensive application [27].

## 3. Biochar-Derived Free Radicals

As previously mentioned, BC has a broad-prospective use in the treatment of environmental xenobiotics, in soil amendment, in photocatalytic and photothermal systems, for photothermal conversion, as electrical and thermal devices, and as 3D solar vapor-generation devices for water desalination [118,119,120,121]. All these potential uses are due to its high surface area and rich pore structure, which provide great physical absorptivity. They also depend on the chemical characteristics of BC, including the presence of PFRs [122,123]. In this regard, it is of paramount importance to clarify the formation mechanism of free radicals in BC for the optimal management of their properties and their more efficient and safer utilization [124].

### 3.1. Persistent Free Radicals (PFRs)

An atom or molecule with at least a lone pair of electrons is a chemical species characterized by significant instability and high chemical activity and is referred to as a free radical species [107]. Usually, free radicals are highly unstable and rapidly react with each other, thus being destroyed as soon as they form, with a consequent very short half-life. However, it has been found that in BC, some free radicals, named persistent free radicals (PFRs), like the radicals that naturally occur in the environment, known as environmental persistent free radicals (EPFRs), can remain stable for months and play a crucial role in the subsequent reactions of oxidative degradation carried out by BC containing them [25,107,125,126] (Figure 6).

Unlike other free radicals, PFRs are resonance-stabilized since they are bound to the external or internal surface of solid particles of BC. They can be analyzed by electron paramagnetic resonance spectroscopy (EPR) [25]. Figure 7a provides an example of an EPR analysis of the PFRs present on a solid N-doped hydro char prepared in a tube furnace at a temperature of up to 600 °C for 1 h under a N_2_ atmosphere [127].

Their lifetime under a vacuum appears infinite, while they react with molecular oxygen in the air, resulting in decay with time and the simultaneous production of reactive oxygen species (ROS). In this regard, PFRs act as transition metals like Fe^2+^, stimulating ROS production in aqueous systems. Unlike PFRs, ROS are detectable by EPR only when captured by a proper radical scavenger, such as 5,5-dimethyl-1-pyrroline-N-oxide (DMPO). Figure 7b provides an example of the EPR spectrum of the unstable free radical superoxide (O_2_•^−^) when trapped by DMPO to form a long-lived nitroxide radical (DMPO-OOH). In BC analyzed using the EPR technique, PFRs were previously detected in combustion-generated particulate matter (PM), sediments, and soils. PFRs can be categorized into three classes, i.e., oxygen-centered PFRs (OCPFRs), carbon-centered PFRs (CCPFRs), and oxygenated carbon-centered radicals (CCPFRs-O). The EPR analyses provide three parameters: the PFR concentration, the g-value, and the line width [107] (Figure 8).

PFR concentration is calculated from the double integral of the EPR spectrum and can reflect the content of PFRs in BC [128]. The g-value of PFRs is a constant specific to a particular compound, reflects its hybrid nature, and provides information about the type of radical [129]. The PFR line width in the EPR spectrum measures the peak-to-peak width. This is affected by spin–spin interactions (including electron–proton interaction and electron–electron interaction), the heteroatom effect, and the anisotropy of the spectrum [107].

The line width reflects the relaxation time of spinning electrons [130]. It has been reported that the oxidation processes that can occur using BCs mainly depend on PFRs and these parameters [125,131,132]. These parameters are, in turn, affected significantly by pyrolysis conditions, biomass types, the elemental composition of pristine biomass, and the presence of external transition metals (Table 6).

Qin et al. [132] found that the PFR concentrations in the same BC that were obtained at different temperatures and those in different kinds of BC obtained at the same temperature were significantly different. Tao et al. [135], as well as Xiang [136] and Huang et al. [24], found that in BCs from different feedstocks, the PFR concentrations first increased with increasing temperature, reaching a maximum around 500–600 °C, and then decreased with a further increase in temperature. The relations between the feedstocks’ properties or the BCs’ composition with the PFR concentrations were also demonstrated [133], and non-lignocellulosic-biomass produced lower PFR concentrations than lignocellulosic-biomass under the same pyrolytic conditions, perhaps due to their lower H/C and O/C atomic ratios [134]. The types of PFRs that can be produced during pyrolysis change in the pyrolysis process and change along with a temperature rise, as reported in Table 6. One study speculated that the reduction in oxygen content during biomass pyrolysis may account for the progressive conversion of oxygen-centered radicals to carbon-centered radicals [134].

#### Mechanism Proposed for PFR Formation during Biomass Pyrolysis

The several environmental sources of PFRs include atmospheric particulate matter (PM), contaminated soil, materials from thermal treatments of plastic and hazardous waste, tar balls, and the products deriving from the pyrolysis of biodiesel and biomass waste feedstocks at high temperatures [25]. Concerning BC-derived PFRs, it was observed that they mainly form in the post-flame and cool-zone regions of combustion systems and other thermal conversion processes. Although the actual mechanism by which PFRs form during pyrolysis remains not fully clarified, transition metals capable of electron transfer and the substituted aromatics molecules present in lignin have been recognized as critical factors of PFR formation. However, high concentrations of PFRs have also been detected in the product of combustion of non-aromatic cellulose in the absence of transition metals [135]. Based on the temperature of pyrolysis processes, during the production of BC, highly heterogeneous composite structures occur, comprising both labile and recalcitrant organic molecules, such as PAH, furans, and dioxins, as well as inorganic fractions including oxides, cations, anions, and free radicals [137]. These fractions, products of the incomplete combustion of biomass, may gradually form PFRs by different pathways, including or not including transition metals. The formed PFRs could be either only surface-stabilized or be surface-stabilized in metal-radical complexes [27]. Generally, the breaking of covalent bonds by heat, light, electricity, and chemical energy, is essential to form free radicals; during the pyrolysis process of lignocellulosic biomasses. Their main constituents, namely, cellulose, hemicellulose, and lignin, undergo different reaction pathways at various destructive pyrolysis temperatures of 300 °C, 300–400 °C, and 350–450 °C. Anyway, the presence of transition metals can strongly affect the possible formation of PFRs during pyrolysis. Figure 9 attempts to describe the possible series of events occurring during biomass pyrolysis that could lead to the formation of PFRs, which are also chemically described in Figure 1 (concerning lignin) and Figure 2 (concerning cellulose and hemicellulose).

First, the C-O and C-C covalent bonds of constituents of lignin are broken under heat, either via electron transfer by transition metals or not, to form free radical fragments, phenols, chinones, and other products of incomplete combustion. Simultaneously, the cleavage of the glucoside bonds of cellulose and hemicellulose that are present in biomass feedstocks occurs, causing depolymerization and the formation of other radicals. These first radicals can couple to form bio-oil, pyrolysis syngas (CO_2_, CO, CH_3_CH_3_, and CH_4_), and BC simultaneously or may abstract hydrogen from other molecules, forming further radicals [133,138]. Several chemical reactions can occur, including dehydration, decarboxylation with further emissions of CO_2_, CO, and H_2_O, aromatization, and intra-molecular condensation leading to the formation of the crystalline graphene structure and graphitic radicals. During pyrolysis, the elemental composition of biomass undergoes changes that cause mutations in the types of radicals that, upon entrapment onto the BCs’ surface and/or the formation of metal–radical complexes, form stable PFRs [24].

According to findings reported in the literature, the possible types of PFRs comprise (i) transition metal-mediated PFRs or (ii) PFRs forming inside organic matrices during biomass pyrolysis to give BC [139]. The transition metal-mediated PFR formation starts with the initial physisorption of an aromatic substituted molecule or of its degradation intermediate radicals, generated at 150–400 °C or under UV irradiation onto transition metal oxides such as ZnO, NiO, CuO, Fe_2_O_3_, and TiO_2_ or transition metal ions [140]. Then, chemisorption occurs by forming a chemical bond eliminating water or hydrogen chloride. Finally, a single electron is transferred from the substituted aromatics to the center of the transition metals, leading to the simultaneous reduction of metal and the formation of PFRs [140], the stability of which can be attributed to the synergy of metals and aromatic compounds [140]. A transition metal accepts an electron, and its valence changes from high to low during this process.

Unlike the PFRs discussed previously, PFRs formed inside the matrix of organic moieties are not related to the presence of transition metals [139]. Still, they are highly dependent on the relevant organic matter, while their concentration is significantly and positively correlated with the elemental carbon content [139]. In this case, PFRs are compared in terms of thermally treated particles, for which the breaking of chemical bonds in the precursor molecules during pyrolysis is the primary reason. At the initial pyrolysis stage, the homolytic cleavage of weak linkage bonds like the α- and β-alkyl aryl ether bonds, C-C, and C-O linkage resulted in the forming of free radicals in BC. The outer-surface free radicals would rapidly react and dissipate, resulting in a decrease in EPR signals. The free radical concentrations then increased with extended pyrolysis and during the cooling stage, thus accumulating many free radicals on the BC’s surface [139] and dramatically boosting the EPR signals. The free radicals formed in the matrix of the produced BC are probably protected from reacting with each other or other chemicals and are thus stabilized.

As mentioned earlier, the type of biomass and its elemental composition, the presence of oxygenated functional groups, the pyrolysis conditions (temperature, heating time, and heating rate), and the presence of external transition metal as well as phenolic compounds strongly affect both the concentration, structure, and type of PFRs present in BC. Notably, no radical is produced during the first stage of pyrolysis, providing the transition char (< 300 °C). Subsequently, in the second stage of pyrolysis (300–500 °C), amorphous char is produced, and oxygen-centered radicals and oxygenated carbon-centered radicals appear. In the third stage of pyrolysis at 500–700 °C, composite char is created, wherein the concentrations of PFRs, including carbon-centered and oxygenated carbon-centered radicals, drastically decrease. Finally, when turbostratic char is produced (> 700 °C), little or no PFRs are subsequently produced [27]. In the EPR, the g-factor values, even if they could change due to the presence of metal ions and temperature changes, are specific for a type of radical. Table 7 reports the main types of radicals recognizable in BC and their specific g-values.

### 3.2. PFRs: Light and Shadows

#### 3.2.1. PFR Light

It has been demonstrated that PFRs originating in BC by combustion in the presence or absence of external transition metals could play a vital role in several beneficial reactions, such as the PFR-mediated remediation and degradation of organic and inorganic pollutants by different actions and mechanisms, including oxidative and reductive processes (Table 8).

For instance, PFRs on BC can activate hydrogen peroxide (H_2_O_2_) or oxygen (O_2_), as well as persulfate (S_2_O_8_^2−^), to produce different free oxygenated radicals (ROS) that are capable of efficiently degrading organic contaminants such as chloro-biphenyl [32], phenolic compounds and polychlorinated biphenyls [21], diethyl phthalate [33], thiacloprid [123], and bisphenol A [25]. Moreover, organic chemicals can also be directly degraded on the BC surface by macromolecular free radicals without adding any radical activators [74]. The semiquinone-type radicals present in BC can oxidize As (III) [142]. At the same time, BCs can also exhibit the highly effective removal of Cr (VI) by reduction to Cr (III) using PFRs for industrial wastewater remediation [143,144,145,146,147].

Unfortunately, PFRs, by generating surface-bound hydroxyl radicals and free hydroxyl radicals in aqueous solution and also in the absence of H_2_O_2_, can induce various types of cardiovascular and pulmonary disease through ROS-induced oxidative stress (OS) [25]. PFRs and OH radicals that were detected in biological fluids generated ROS that induced an oxidant injury and modulated toxic responses in biological tissues [149]. Moreover, quinoid redox cycling is another possible path causing the formation of ROS from material containing semiquinone-type radicals, which could exert toxicity like that exercised by the combustion products present in cigarette smoke [150]. Although BC has beneficial effects on agricultural soil, the PFRs in BC could inhibit plant germination and growth when used in soil remediation. BC addition as a soil amendment has been reported to positively affect plant germination, growth, and yield [151,152]. In contrast, a negative impact has also been documented when BC-bounded PFRs induce ROS, which can inhibit seed germination and retard the growth of roots and shoots [32,35]. As shown in Figure 10, the formation and presence of PFRs in the BC produced by several biomasses have been widely documented and studied since 2014.

In this regard, in Table 9, we have reported a random selection of the main experimental works regarding the PFRs found in BCs, obtained by different biomasses conveyed in recent years (2019–2024). Table 9 also summarizes their reported applications, including mainly the oxidative degradation of organic environmental pollutants (51 papers), the removal of hazardous inorganic compounds from wastewater such as As (III) and Cr (VI) (12 papers), the degradation of biological samples, including bacteria (3 papers), hormones (4 works), genes of bacterial resistance (1 paper), and their use as electrical devices due to their electron and electron donor capacities (EAC and EDC).

Figure 11 shows the relative abundances of the types of PFR applications concerning the 72 case studies considered here.

As for the mechanisms, many publications regarded the activation, which was sometimes photocatalytic, of PS, PMS, and PDS by BC. The employed BC was derived from different feedstock biomasses (bagasse powder, poplar and pine sawdust, cellulose, lignin, blue algae, waste straw, and other sources, as reported in Table 9), not doped, or doped with nitrogen atoms or different metals including Fe, Mn, Co, Ni, Zn. In these processes, the electron transfer promoted by metals and/or PFRs of a diverse nature, based on the pyrolysis conditions, generated ROS such as SO_4_•^−^, •OH, •O_2_^−^, •O_2_H and non-radical species (^1^O_2_), which carried out the oxidative degradation of different organic xenobiotics, including drugs, dyes, antibiotics, and hormones, as well as phenols or aromatic derivatives. Many other publications reported the use of BC to activate or photochemically activate O_2_ or H_2_O_2_ (Fenton-like systems) via metal and/or PFRs-mediated single-electron transfer. The generated ROS (•OH, •O_2_^−^, •O_2_H) and oxygen non-radical species (^1^O_2_) successfully oxidized several organic pollutants, degraded hormones, and eDNA and, in some cases, showed antibacterial effects against *E. coli* and *S. aureus*. Moreover, the capacity of BC to transfer electrons via transitional metals or PFRs was used to oxidize As (III) to As (V) or reduce Cr (VI) to Cr (III), thus resulting in helping to remove hazardous inorganic contaminants from industrial wastewater. A notable recent review reported on the efficiency of BC/layered double hydroxide composites as catalysts for the treatment of organic wastewater by advanced oxidation processes [213]. Several studies reported in this paper by Liu et al. demonstrated that degradation processes were based on radical reactions triggered by BC-associated PFRs [213].

#### 3.2.2. BC-Associated PFRs Shadows: Cytotoxicity and Biotoxicity

Despite the plethora of possible beneficial applications of BC, PFRs, as well as other free radicals and the toxic substances that compose BC, such as heavy metals, PAHs, dioxins, and perfluorochemicals (PFCs), are released into the environment during the pyrolysis process, thus representing a potential risk to the environment and humans [214]. Additionally, as well as other contaminants, the possible carbon allotropes formed during pyrolysis are severe contaminants in air, water, and soil [215]. Black carbon, carbon black (CB), carbon nanotubes, graphene, quantum dots, and fullerenes can possess distinct toxicity that depends on many factors, including the type of allotrope, particle size, form, structural defects, coating molecules, and grade of functionalization [215]. Understanding the toxicity of carbon nanomaterials and nanoproducts that are possibly present in BC is essential for human and environmental health, safety, and public acceptance. In this regard, recent studies have focused their attention on the adverse effects of BC due to its particle size and the various interactions with the environment that could occur [216,217]. Upon its application, BC may produce harmful environmental effects due to aging by oxidative or biological processes, leading to changes in its properties [218,219]. Additionally, higher toxicity has been reported for BC with micro- or nano-dimensioned particles. It has been reported that the presence of micro-BC (MBC) or nano-BC (NBC) can promote the release of heavy metal ions into the medium when applied to soil [214]. Kim et al. (2018) observed that BC particles with a particle size of less than 0.45 μm could increase the release and mobility of As in soil [220]. Regarding the biotoxicity of MBC/NBC, it has been previously reported that particle-induced oxidative stress is a crucial mechanism of MBC/NBC cytotoxicity, which increases as the particle size decreases. Also, the PFR concentration on the surface of particles with an aerodynamic diameter of less than 1 μm is the highest [139,221]. While several reviews and studies exist on the production and modification of BC, the reaction mechanisms, and the beneficial active role of BC in environmental remediation, the adverse effects and potential risks of BC have only recently been evidenced. The comprehensive phenomena and mechanisms involved in BC toxicity still require elucidation, especially in environmental media different from soil, including water and the atmosphere. It is imperative to systematically study and discuss the possible adverse environmental effects of BC application concerning various media, including water and the atmosphere, by determining the corresponding occurrence, detection, assessment, and avoidance measures. Worryingly, the current knowledge concerning the possible adverse effects on the environment and biota deriving from the extensive application of PFRs originating in BC is even more limited [222]. Although they are emerging as contaminants of increasing concern, their formation, fate, toxicity, and health risks are poorly known [222]. Thermal treatment, a common remediation technique to clean industrial soils, induces the formation of PFRs; this could paradoxically increase soil toxicity, which is contrary to the original remediation objective. For example, there is still little knowledge on the formation and toxicity of PFRs in soils contaminated by polycyclic aromatic hydrocarbons (PAHs) [223]. BC-derived PFRs, as well as those present in the environment and deriving from combustion and soil restoration, the burning of coal, wood, straw, cigarettes, oil, and other fuels, and from the restoration of organic contaminated soil, can enter the human body mainly through three pathways including the respiratory tract, from skin exposure, and via ingestion [149]. PFRs are not toxic to living beings and the environment, but they can stimulate the formation of other harmful substances and free radicals, including various types of ROS, when in the environment or in vivo [223]. As is well documented, ROS can interfere with the normal redox and metabolic processes, thus causing oxidative stress in biota [224]. Additionally, it has been reported that exposure to PFRs may induce cell degeneration or apoptosis and may affect the normal functions of the heart or lungs of humans [223]. So far, cytotoxicity and biotoxicity are the two categories of toxicity reported as attributable to PFRs (Table 10).

Usually, toxicity tests are carried out in research laboratories using both environmental samples and lab-prepared PFRs, such as those generated by MCP230 (a mixture of CuO and chlorophenol at 230 °C), DCB230 (a mixture of CuO, 0.2 μm amorphous silica and 1,2-dichlorobenzene at 230 °C), CGUFP (combustion-generated ultrafine particle) or other mixtures of transitional metals and substituted aromatic compounds. For cytotoxicity experiments, cultured cells extracted from the bronchial epithelium and rats are used, while biotoxicity essays are carried out on plants, fishes, rabbits, and worms. Generally, it was observed that exposure to PFRs causes oxidative stress. More specifically, the cytotoxicity tests evidenced cell variation with decreased numbers and activity, a disparity in protein expression, and DNA damage. Biotoxicity experiments revealed abnormalities in development and behavior, disease, and organ and tissue damage. Although BC can serve as an environmentally sustainable soil amendment material due to its ability to enhance several chemical properties of soil, such as pH, electrical conductivity, CEC, and organic carbon content, thus contributing to the overall improvement of nutrient retention in the soil, BC amendments with high concentrations of PFRs negatively affect crop growth.

Additionally, it has been found that PFRs used in aquicultural solutions inhibited the germination rate of different crops by ROS induction [149]. The oxidative stress brought about by the production of ROS can also damage the plasma membrane of the root system and hinder plant root growth. Moreover, PFRs induced neurotoxicity in *Cryptobacterium hidradenoma*, transforming it into a neurotoxin for soil organisms and thereby posing a threat to their survival.

## 4. Future Challenges and Risk Prevention Strategies

This review has evidenced that BC and mainly the PFRs generated during the pyrolysis processes performed to produce it could be double-edged weapons. BC is reported to be an eco-friendly and low-cost black gold with many beneficial properties, including the capability to remove organic and inorganic pollutants from water by adsorption processes and/or through its PFRs. However, several studies have reported that PFRs can be very dangerous to the environment and humans through a ROS-dependent mechanism. In addition to being produced by various common xenobiotics, PFRs can easily be converted into secondary pollutants, causing further biotoxicity. The still too-little-studied transport and transformation of PFRs in the environment can also affect the behavior of other substances, leading to potential environmental hazards that are not yet fully understood. Therefore, further exploration of the ecological impact of PFRs and the development of prevention and control measures are necessary. In this regard, although some progress has been made in terms of environmental risk and biotoxicity studies concerning PFRs, research is still in the initial stages, and there is an urgent need for systematic and in-depth studies on the production and transport of PFRs. A more in-depth understanding of the influence of environmental conditions on their occurrence is needed to control the external factors for reducing the negative output of PFRs and promoting their degradative action on xenobiotics. More rational knowledge about their toxicity mechanisms is necessary to have more precise toxicological equivalence regulation. A better strategy to prevent the risks associated with PFRs could be to avoid exposure to them by reducing contact with combustion sources, such as vehicle exhaust and cigarette smoke. Proper air filtration systems removing PFRs from indoor air and wearing protective masks or respirators could lower the possibility of contact with PFRs in outdoor environments. Additionally, treatments for limiting the adverse health effects associated with PFR exposure, such as using antioxidants, which can neutralize ROS, could be another strategy to protect humans from the adverse impacts of PFR exposure.

## 5. Conclusions and Future Perspectives

The pivotal role of BC-related PFRs in BC catalytic efficiency in water and soil pollution remediation has been reviewed in this paper. The main mechanisms by which PFRs can originate and the main methods to detect them on BC have been discussed. The key roles of feedstock biomasses and pyrolysis conditions in the formation of PFRs has also been reported. Several recent case studies concerning the critical role of PFRs in the catalytic oxidative degradation by BC of organic pollutants and in the removal of Cr/As and other metal ions in aqueous phase have been reported and discussed. It has been evidenced that for organic pollutants remediation, PFRs act as activator agents of oxidant substrates, which are subsequently involved in the degradation process. Otherwise, for metal ion removal, the PFRs on BC act as electron transfers for the adsorption and concomitant reduction/oxidation of metals by BC. Finally, PFRs on BC could also help mediate the recycling of Fe^3+^/Fe^2+^ in Fenton-like processes to enhance the efficiency of BC in pollutant removal. However, many aspects remain unclear concerning PFRs, including the influence of BC size on the evolution of PFRs, the exact relationship between the reactivity of BC and its size, and the overall roles and relative significance of PFRs, quinone moieties, and the carbon structure of BC in the activation of oxidizing agents and in the redox transformation of inorganic contaminants. Furthermore, while several studies about the laboratory applications of BC for wastewater remediation have been developed, future studies should concentrate on the up-scaled utilization of BC for the treatment of different real wastewaters, such as industrial wastewater and municipal wastewater. To this end, efforts are needed to design proper reactors and to develop methods for the large-scale production of the desired BC. Finally, to better contribute to the circular economy, the reutilization of spent BC should be given serious consideration.

## Data Availability

Not applicable.

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
