# Peer review of "Biochar-Derived Persistent Free Radicals: A Plethora of Environmental Applications in a Light and Shadows Scenario"

_toxics, 2024, doi:10.3390/toxics12040245_

Round 1

Reviewer 1 Report

Comments and Suggestions for Authors

The submitted manuscript represents information about biochar-derived persistent free radicals.

The topic is interesting and important. However, in my opinion, the authors devote too much space to the already well-known and often described properties of biochar and their dependence on the conditions of the pyrolysis process, the type of raw material, etc. Of course, these issues are important, but they should be discussed only in the context of PFRs. Manuscript fits well into the journal scope. Minor revision are necessary in the all sections. 

Comments are provided below.

L35. Please add more references. Applications of biochar is a very wide and important issue.

L38. ..heavy metals, dyes, or phosphate from aqueous solutions.. The possibilities of using biochar as an adsorbent are much wider i.e pharmaceuticals,  fulvic acids, pesticides, PAHs, petroleum derivatives, more ions. Please add also more references.

L79. What about PFR’s in biochars obtained from waste materials such as sewage sludge, biogas residues, kitchen waste, etc.?

Table 1. Main sources and general application of BCs. The compilation of information is ok, but one reference is not enough. The comment also applies to other tables included in the manuscript

Author Response

The submitted manuscript represents information about biochar-derived persistent free radicals.

The topic is interesting and important. However, in my opinion, the authors devote too much space to the already well-known and often described properties of biochar and their dependence on the conditions of the pyrolysis process, the type of raw material, etc. Of course, these issues are important, but they should be discussed only in the context of PFRs. Manuscript fits well into the journal scope. Minor revision are necessary in the all sections.

We thank the Reviewer for his/her positive comments and for the work of revision made for us. In our opinion, although the main scope of this paper was to review BC-derived PFRs and their application, we thought that providing readers an as complete as possible background on BC, its production methods and its properties, before introducing PFRs, could have attracted the attention of a wider audience, including those readers not fully expert in this sector.

Comments are provided below.

L35. Please add more references. Applications of biochar is a very wide and important issue.

As asked by the Reviewer, more sentences and more references on BC applications have been included. Please, see lines 41-52 and new Refs. 3-11.

L38. ..heavy metals, dyes, or phosphate from aqueous solutions.. The possibilities of using biochar as an adsorbent are much wider i.e pharmaceuticals,  fulvic acids, pesticides, PAHs, petroleum derivatives, more ions. Please add also more references.

As asked, additional applications of BC as an adsorbent, including those suggested by the Reviewer have been inserted at lines 56-59, with the new related references (Refs. 13-17).

L79. What about PFR’s in biochars obtained from waste materials such as sewage sludge, biogas residues, kitchen waste, etc.?

The required information has been added. Please, see lines 104-106 and the related new reference 41.

Table 1. Main sources and general application of BCs. The compilation of information is ok, but one reference is not enough. The comment also applies to other tables included in the manuscript.

We apologize in advance with the Reviewer, but in our opinion a reference for each item is sufficient, also because in the subsequent sections all information reported in Table 1 (Table S1 in the revised version) has been more in deep discussed reporting additional references. Moreover, as commented by the same Reviewer, we have already devoted too much space at information on biochar not dealing with PFRs.

Reviewer 2 Report

Comments and Suggestions for Authors

Biochar (BC) is a carbonaceous material obtained by pyrolysis at 2001000 °C in the limited presence of O2 of different vegetable and animal biomass feedstocks. BC has demonstrated great potential, mainly in environmental applications, due to its high sorption ability and persistent free radicals (PFRs) content. These characteristics enable BCs to carry out the direct and PFRs-mediated removal/degradation of environmental organic and inorganic contaminants. This manuscript reported biochar-derived persistent free radicals for environmental applications. Specific comments are shown below to improve the manuscript.

“we have first reviewed the most common methods used to produce BC, its main environmental applications, the primary mechanisms by which BC remove xenobiotics, as well as the reported mechanisms for PFRs formation in BCs.” Several review papers have been reported about biochar-derived persistent free radicals for environmental applications. The authors should summarize published reviews about the topic, and point out why this review is essential.

The section of “2. Biochar (BC)” should be refined, because biochar production, characterizations and application has been summarized bby many review papers.

“2.2.1. The Question of Temperature” the subsection is confused because no 2.2.2.

“Table 1. Main sources and general application of BCs”, the table provides less information, and thus can be removed or put in Supporting Information.

“Table 4. Techniques typically used to characterize BCs in terms of their physicochemical, surface, 193 and structural characterization” can be removed or put in Supporting Information.

References are suggested to support this work: Degradation of tetracycline by nitrogen-doped biochar as a peroxydisulfate activator: Nitrogen doping pattern and non-radical mechanism. Sustainable Horizons, (2024). 10, 100091; Biochar/layered double hydroxides composites as catalysts for treatment of organic wastewater by advanced oxidation processes: A review. Environmental Research, (2023). 116534.

The tables and figures obtained from literature should request permission form the publishers.

“4. Risk Prevention Strategies and Conclusions” should be separated into two sections. Moreover, a section describing challenges and perspectives should be added.

Grammar and typewriting errors should be checked and corrected in the manuscript.

Comments on the Quality of English Language

Minor revisions.

Author Response

Biochar (BC) is a carbonaceous material obtained by pyrolysis at 200−1000 °C in the limited presence of O2 of different vegetable and animal biomass feedstocks. BC has demonstrated great potential, mainly in environmental applications, due to its high sorption ability and persistent free radicals (PFRs) content. These characteristics enable BCs to carry out the direct and PFRs-mediated removal/degradation of environmental organic and inorganic contaminants. This manuscript reported biochar-derived persistent free radicals for environmental applications. Specific comments are shown below to improve the manuscript.

“we have first reviewed the most common methods used to produce BC, its main environmental applications, the primary mechanisms by which BC remove xenobiotics, as well as the reported mechanisms for PFRs formation in BCs.” Several review papers have been reported about biochar-derived persistent free radicals for environmental applications. The authors should summarize published reviews about the topic, and point out why this review is essential.

We thank the Reviewer for his/her suggestion which has enabled us to better highlight the relevance of our work. To address the Reviewer request we have included the following paragraphs at the end of Introduction Section (lines 133-146):

“To confirm the relevance and essentiality of the present paper, a recent survey in PubMed data set has evidenced that, although the number of studies on BC-related PFRs are increasing in the last years, they are still very limited if compared to those on BC in general (134 vs 11646 from year 2014 so far). Additionally, the review articles on BC-associated PFRs, that by gathering information on the topic, could stimulate more research on it are indeed limited (16). Some recent examples could be the works by Zhang et al, Liu et al, Luo et al and Yi et al [1,44-46]. On this scenario, this review can be considered essential, because it offers an all-round and complete overview both on BC and BC-related PFRs, via an extensive discussion on both their beneficial impact and the possible risks to humans and the environment that could derive by their widespread and irrational application. With the present paper, we have provided readers a reader-friendly work, where the information has mostly been organized into easy-to-read Tables, Schemes and statistical graphs which could have an excellent impact on the audience.”

The section of “2. Biochar (BC)” should be refined, because biochar production, characterizations and application has been summarized by many review papers.

As asked by the Reviewer, Section 2 has been shortened and refined, mainly by moving some Tables in the new Supplementary Materials file.

“2.2.1. The Question of Temperature” the subsection is confused because no “2.2.2”.

In our design, since pyrolysis temperature strongly influence the BC features and properties, “The Question of Temperature” Section should be a Subsection of 2.2. Section, which reports the BC characteristics. In this regard, being “The Question of Temperature”, the first subsection of Section 2.2., it should be indicated with the numbering 2.2.1., as in the original paper.

“Table 1. Main sources and general application of BCs”, the table provides less information, and thus can be removed or put in Supporting Information.

As asked, Table 1 has been moved in the new Supplementary Materials file.

“Table 4. Techniques typically used to characterize BCs in terms of their physicochemical, surface, 193 and structural characterization” can be removed or put in Supporting Information.

As asked, Table 4 has been moved in the new Supplementary Materials file.

References are suggested to support this work: Degradation of tetracycline by nitrogen-doped biochar as a peroxydisulfate activator: Nitrogen doping pattern and non-radical mechanism. Sustainable Horizons, (2024). 10, 100091; Biochar/layered double hydroxides composites as catalysts for treatment of organic wastewater by advanced oxidation processes: A review. Environmental Research, (2023). 116534.

The suggested references have been included in the revised manuscript as Ref. 156 (Table 9) and Ref. 213 (lines 679-683).

The tables and figures obtained from literature should request permission form the publishers.

Except for Figure 7, no other Figure in our work has been obtained from reported works, and don’t need permissions. Concerning Figure 7, we have request and obtained the license of reproduction by Elsevier (now reported in the Figure caption, lines 456-457). Furthermore, Tables have been originally constructed by us, using data in literature. The works sources of such data have been cited.

“4. Risk Prevention Strategies and Conclusions” should be separated into two sections. Moreover, a section describing challenges and perspectives should be added.

As asked by the Reviewer, Conclusions have been inserted in a new section (lines 791-803), while challenges and perspectives have been included both in Section 4 (lines 764-783) and in Section 5 (803-813).

Grammar and typewriting errors should be checked and corrected in the manuscript.

Comments on the Quality of English Language

Minor revisions.

The English language of all paper has been checked to improve the quality of the manuscript.